# Facing the Green Threat: A Water Flea’s Defenses against a Carnivorous Plant

**DOI:** 10.3390/ijms23126474

**Published:** 2022-06-09

**Authors:** Sebastian Kruppert, Martin Horstmann, Linda C. Weiss, Elena Konopka, Nadja Kubitza, Simon Poppinga, Anna S. Westermeier, Thomas Speck, Ralph Tollrian

**Affiliations:** 1Department of Animal Ecology, Evolution & Biodiversity, Ruhr-University Bochum, 44780 Bochum, Germany; martin.horstmann@rub.de (M.H.); linda.weiss@rub.de (L.C.W.); elenabarmaeva@live.com (E.K.); nadja.kubitza@rub.de (N.K.); tollrian@rub.de (R.T.); 2Friday Harbor Laboratories, University of Washington, Friday Harbor, WA 98250, USA; 3Plant Biomechanics Group, Botanic Garden, University of Freiburg, 79104 Freiburg im Breisgau, Germany; simon.poppinga@tu-darmstadt.de (S.P.); anna.westermeier@biologie.uni-freiburg.de (A.S.W.); thomas.speck@biologie.uni-freiburg.de (T.S.); 4Freiburg Materials Research Center (FMF), Stefan-Meier-Straße 21, University of Freiburg, 79104 Freiburg im Breisgau, Germany; 5Botanical Garden, Department of Biology, Schnittspahnstraße 2, Technical University of Darmstadt, 64287 Darmstadt, Germany; 6Freiburg Center for Interactive Materials and Bioinspired Technologies (FIT), Georges-Köhler-Allee 105, University of Freiburg, 79110 Freiburg im Breisgau, Germany

**Keywords:** Daphnia, inducible defenses, carnivorous plant

## Abstract

Every ecosystem shows multiple levels of species interactions, which are often difficult to isolate and to classify regarding their specific nature. For most of the observed interactions, it comes down to either competition or consumption. The modes of consumption are various and defined by the nature of the consumed organism, e.g., carnivory, herbivory, as well as the extent of the consumption, e.g., grazing, parasitism. While the majority of consumers are animals, carnivorous plants can also pose a threat to arthropods. Water fleas of the family Daphniidae are keystone species in many lentic ecosystems. As most abundant filter feeders, they link the primary production to higher trophic levels. As a response to the high predatory pressures, water fleas have evolved various inducible defenses against animal predators. Here we show the first example, to our knowledge, in *Ceriodaphnia dubia* of such inducible defenses of an animal against a coexisting plant predator, i.e., the carnivorous bladderwort (*Utricularia x neglecta* Lehm, Lentibulariaceae). When the bladderwort is present, *C. dubia* shows changes in morphology, life history and behavior. While the morphological and behavioral adaptations improve *C. dubia*’s survival rate in the presence of this predator, the life-history parameters likely reflect trade-offs for the defense.

## 1. Introduction

Members of the crustacean family Daphniidae represent some of the most abundant zooplankters in lentic freshwater ecosystems [1]. As consumers of phytoplankton, they link primary production to higher trophic levels by falling prey to a variety of predators such as other crustaceans, fish, or insects [2,3]. This seasonally variable predation risk favored the evolution of inducible defenses in some species of the Daphniidae. Inducible defenses are a form of phenotypic plasticity that decreases an organism’s vulnerability to specific predators (for reviews, see [4,5,6,7]). These defenses range from alterations in morphology or life-history parameters to behavior. Many defenses are predator-specific and adapted to counter the respective predator. For example, a vast diversity of striking morphological defenses has been described. They include rather minute structures such as the ‘neckteeth’ expressed by *Daphnia pulex* [8,9,10], the medium sized ‘crown of thorns’ in *D. atkinsoni* [11], and large morphological changes such as the helmets of *D. cucullata* and *D. lumholtzi* [12,13], as well as the crests of *D. longicephala* [14,15,16]. Further, alterations in the carapace architecture and its mechanical properties have been reported [17,18,19]. In the presence of visual hunters like fish, some *Daphnia* species alter their life history and shift resources from somatic growth to reproduction [20,21]. Species of *Daphnia* predated by invertebrates display the opposite strategy by accelerating their somatic growth. This way, the prey overcomes the predator’s gape limit, at the expense of population growth rate [22,23,24,25]. A well-studied behavioral defense strategy is the diel vertical migration, avoiding visual predators by residing in deeper waters during the day and ascending into shallow nutrient-rich strata for grazing during the night [26]. In addition, changes in swimming behavior have been reported, like the predator-induced increase or decrease in individual swimming speed (for a review, see [27]).

Existing literature on inducible defenses in Daphniidae focusses on responses against animal predators, and carnivorous plants have been overlooked in this regard so far. We found but a single study on *Daphnia* inducible defenses that took carnivorous plants into account, but reported no phenotypic response to *Utricularia* presence [28].

The bladderwort (*Utricularia x neglecta* Lehm; formerly known as *U. australis*), an aquatic carnivorous plant native to Central Europe, is a naturally coexisting predator of many different *Daphnia* species including *Ceriodaphnia dubia* [28,29,30]. With its ultrafast suction traps, it can catch its prey within ~5 ms, leaving little to no chance of escape [31]. Water is actively pumped out of the trap lumen via specialized glands [32], creating a sub-ambient pressure [33,34,35]. If triggered, the prey (*C. dubia*) is sucked into the trap with a speed of up to 4 m/s [31]. The trap resets in about 15–30 min after suction and continues to catch further prey until the trap is full. With these highly efficient traps, the plant acquires a substantial nutrient supply [31,35,36,37]. Many daphniids fit into the suction traps and are therefore potential prey [38]. In combination with seasonally high abundances of *U. x neglecta*, and due to the fact that each plant can possess several hundreds to thousands of traps, it may pose a severe threat to daphniid populations, which constitute a substantial portion of the prey [30,31,39]. In this context, we hypothesized that *Daphnia* may have evolved mechanisms to reduce this predation pressure. 

Following our hypothesis, we designed experiments to answer the following questions: (1) Does the presence of *U. x neglecta* induce morphological alterations in *C. dubia*? (2) Does the presence of *U. x neglecta* induce life-history alterations in *C. dubia*? (3) Does the presence of *U. x neglecta* induce alterations in phototaxis or swimming habit (i.e., behavior) in *C. dubia*? (4) Do the observed alterations in *C. dubia* reduce the predation efficiency of *U. x neglecta*?

Using high-resolution 3D morphometrics [40], we investigated *C. dubia* for morphological changes as adaptive responses to the presence of *U. x neglecta*. Additionally, we analyzed life-history shifts and behavioral alterations as a possible response of *C. dubia* to the plant’s presence. Furthermore, we analyzed the bladderwort’s capture efficiency of exposed (defense-induced) and naïve (uninduced) *C. dubia* in order to determine the protective effect of the displayed defensive strategies. 

## 2. Results

### 2.1. Trap Entrance Dimensions

The *U. x neglecta* trap entrance dimensions were determined as 495 µm (±166 µm SD) average height and 613 µm (±147 µm SD) average width (*n* = 20 each). Therefore, the trap entrances are typically wider than they are high (ratio ~1:1.23).

### 2.2. 2D Investigation

We found a significant effect of time and treatment on *C. dubia*’s body lengths as well as a significant interaction (MANOVA; time: *F* = 437.163, *DF* = 6, *p* < 0.001, η^2^ = 0.606; treatment: *F* = 114.530, *DF* = 3, *p* < 0.001, η^2^ = 0.079; treatment × time: *F* = 5.367, *DF* = 17, *p* < 0.001, η^2^ = 0.021) (Figure 1A, Appendix A). The same holds true for our analysis of the animals’ normalized body width (time: *F* = 253.224, *DF* = 6, *p* < 0.001, η^2^ = 0.508; treatment: *F* = 41.110, *DF* = 3, *p* < 0.001, η^2^ = 0.041; treatment × time: *F* = 4.789, *DF* = 17, *p* < 0.001, η^2^ = 0.027) (Figure 1B, Appendix A). As the time effect represents mere growth, we focused our analysis on the differences between treatments within the individual days. Here, the induced animals showed significant differences in comparison to the controls. The body lengths of the control animals were significantly higher than those of animals in the ‘fed *Utricularia*’ treatment from day 3 onwards. This pattern strengthened until, from day 5 onwards, both of the control treatments were significantly taller than both of the induced treatments (Appendix A). Similarly, but not as pronounced, the normalized body widths were larger in the control treatments than in the induced group. This difference became visible in the ‘tap water control’ from day 4 onwards and strengthened until, on day 6, the pattern was similar to that observed in the body lengths (Appendix A). 

### 2.3. 3D Analysis

Using the approach described by Horstmann et al. [40], we confirmed the same significant differences between the control (Figure 2A) and the *Utricularia*-exposed animals in the five-day-old specimens (Figure 2B). These differences in overall appearance (Figure 2C) are supported by the confidence ellipsoid analysis, as it revealed no overlaps, indicating the overall difference between both morphotypes (Figure 2D). We found mean Pearson’s *r* effect sizes of 0.733 for the dorso–ventral body axis, 0.791 for the anterior–posterior body axis, and 0.556 for the lateral body axis. The *Utricularia*-exposed animals (Figure 2B) were smaller than control animals (Figure 2A) of the same age (ctrl = 0.725 ± 0.0175 mm, induced = 0.529 ± 0.038 mm, reduction of 27%). Additionally, we found that the *Utricularia*-exposed animals were slimmer than the controls (Figure 2E). The landmarks of the dorsal and ventral region were significantly shifted towards the anterior–posterior body axis (Wilcoxon tests and FDR testing: *p* < 0.01, *q* < 0.001, Figure 2H). Along the anterior–posterior body axis (Figure 2F), the landmarks were significantly shifted towards the dorso–ventral body axis (*p* < 0.01, *q* < 0.001, Figure 2I). In a lateral direction, the strongest altered regions were the head, neck, and brood pouch (Figure 2G). The head’s lateral width was larger by about 90 µm (37%), leading to a total lateral width of 365 µm. The neck region’s width was larger by about 120 µm (35%), leading to a total width of 475 µm. This is mostly due to pronounced fornices that were only visibly formed in *Utricularia*-exposed animals. In the region of the brood pouch, the *Utricularia*-exposed animals were thinner by about 90–120 µm. While these landmark shifts in lateral dimension were proven significant (*p* < 0.01, *q* < 0.001, Figure 2K), the reduction of lateral width in *Utricularia*-exposed animals in the region of the second antenna joint could not be statistically supported (*p* > 0.05, *q* > 0.01, Figure 2K).

### 2.4. Life-History Shifts

Kruskal–Wallis tests revealed no significant differences in the number of egg-carrying females between the treatments (Figure 3, Appendix A). However, from day 4 onwards, the ‘tap water control’ treatment showed a significantly larger clutch size than both *Utricularia*-exposed treatments (day 4: χ^2^ = 317.54, *DF* = 3, *p*-value < 0.001, η^2^ = 0.157; day 5: χ^2^ = 586.6, *DF* = 3, *p*-value < 0.001, η^2^ = 0.119). On day 6, the ‘*Ceratophyllum* control’ treatment also deposited significantly more eggs than the *Utricularia*-exposed treatments (Figure 3, Appendix A) (day 6: χ^2^ = 418.64, df = 3, *p*-value < 0.001, η^2^ = 0.137). 

### 2.5. Behavioral Alterations

#### 2.5.1. Predator Avoidance

During the experiment, the majority of the animals were observed to aggregate in the two upper edges of the tank (Figure 4; ANOVA, F = 5.265, *p* < 0.001, η^2^ = 0.016), i.e., the top sections (Figure 4) of the most left and right columns (Figure 4A). This was true for all treatments. In the control treatments (tap water controls as well as control animals facing *E. canadensis*), no side preference was observed. In all treatments including *U. x neglecta* (*Utricularia*-exposed animals as well as control animals), a significant side preference away from the plant and towards the water surface was observed (Figure 4; ANOVA, *F* = 10.260, *p* < 0.001. η^2^ = 0.036). 

#### 2.5.2. Swimming Modes

We found significant differences in swimming modes between the treatments (Kruskal–Wallis rank sum test; *chi-squared* = 53.978, *DF* = 3, *p* ≤ 0.001, η^2^_Hop and Sink_ = 0.028, η^2^_Zooming_ = 0.023). In the ‘tap water control’, there was no significant difference between the percentages of duration of ‘zooming’ and ‘hop and sink’ swimming modes (Bonferroni-corrected pairwise Wilcoxon test: *p* ≤ 0.05). The animals of the ‘fed *Utricularia*’ treatment showed significant differences in the duration of the used swimming mode (Bonferroni-corrected pairwise Wilcoxon test: *p* ≤ 0.001). They performed ‘zooming’ roughly 25% and ‘hop and sink’ about 75% of the time (Figure 5A). This resulted in significant differences in swimming mode duration between the treatments (Bonferroni-corrected pairwise Wilcoxon test: *p ≤* 0.05 for both comparisons).

#### 2.5.3. Swimming Velocity

Our analysis of the average swimming velocities showed that the induced animals swam significantly slower than the control animals (Kruskal–Wallis rank sum test; *chi-squared* = 359.09, *DF* = 3, *p* ≤ 0.001, η^2^_Hop and Sink_ = 0.06, η^2^_Zooming_ = 0.025, Figure 5B). That was true for both swimming modes (pairwise Wilcoxon test: *p*_hop& sink_ ≤ 0.001; *p*_zooming_ ≤ 0.001). 

### 2.6. Predation Experiments

The analysis of the predation data revealed a significant higher survival rate of induced animals compared to controls (Mann–Whitney U test: U = 10.5, *p* ≤ 0.05, *n* = 10, r = 0.691, Figure 5C). *U. x neglecta* caught every 10th control animal (90% survival), but only every 40th induced animal (97.5% survival).

## 3. Discussion

In this study, we observed predator-induced, phenotypically plastic responses in the form of morphological, life-history, and behavioral traits of *C. dubia* exposed to *U. x neglecta*. The two species, both representatives of cosmopolitan clades, are members of a naturally co-occurring predator–prey system native to Central Europe [28,29,30]. Phenotypic plasticity in plant-animal interactions has long been known and especially herbivore-induced plant defenses are well studied [41]. Furthermore, herbivores are described to express dietary-induced plasticity in morphology and behavior, allowing them to deal with plant defenses [42,43]. However, to our knowledge, plant-induced defenses in an animal have not been described yet. A study by Havel and Dodson also checked for predator responses in a *Daphnia* species using an undetermined *Utricularia* species, but did not report any induced alterations [44]. In the following subsections, we discuss our observed plastic responses and their adaptive benefit together with first insights into the nature and origin of the eliciting cue(s).

### 3.1. Morphological Adaptations

We observed a change in the overall body shape in *C. dubia* when exposed to *U. x neglecta*: the animals were shorter and slimmer (dorsoventrally) but increased their lateral size substantially (37%) via the elongation of their fornices. Given the apparent gape limitation of the bladderwort suction traps, we hypothesize that the defensive mechanism is a combination of functional size increase and suction force reduction at the same time. The elongated fornices can hinder the animal’s entry into the trap by interfering with the trap’s opening, while the slim body simultaneously allows the surrounding water to freely flow into the trap and eventually equalize the pressure difference. We assume that the latter is key to this defense strategy, since an overall increase in body dimensions would lead to a total or near-total blockage of the trap entry, with the result that the animal’s body would experience the (nearly) full amount of the potentially lethal suction forces of the trap. Based on our data, induced animals will only be able to block the smaller trap entrances (lateral dimension: 475 µm; smallest trap entrances: 495 ± 166 µm). As inducible morphological defenses in daphniid species are known to continuously grow with every molting cycle, we are certain that our data merely represent the threshold of the defensive effect and with continuous molts the defensive effect will increase. Additional to the aforementioned blocking effect, the slimmer body may reduce the chances of mechanically triggering the traps. Smaller animals may also face smaller drag forces, which could increase survival chances by reducing the acceleration of the animal towards the trap once the trap is triggered. Any of these effects may also explain the prey preference towards larger prey, as reported for two other *Utricularia* species by Guiral and Rougier [45].

### 3.2. Life-History Adaptations

*U. x neglecta*-exposed animals produced significantly fewer offspring per brood. Such a reduced number of offspring has been reported for *D. magna* as a defense against visually hunting fish that will detect brood carrying females more easily [46,47]. If *C. dubia* similarly produces fewer, but larger offspring that reach a defended stage earlier, the observed reduction in offspring may be a defense as well. However, this remains to be tested. In the case of the mechanosensory-dependent predation by *U. x neglecta*, it is more likely to represent the costs associated with the expression of defenses and the material required for the elongated fornices, and/or of a smaller brood pouch caused by the shape alteration. The decreased somatic growth rate may limit the amount of food that can be ingested, since a reduced body size also limits the food filtration. Therefore, the observations may also be explained by the size-efficiency hypothesis [48].

### 3.3. Behavioral Adaptations

In comparison to morphological and life-history adaptations that require some time to be expressed (here up to 5 days) [10,15,49,50,51], behavioral responses can be expressed quickly [6,27]. Behavioral defenses, especially in *Daphnia*, can therefore function as temporary defenses that bridge the time lag between predator perception and morphological defense expression [52]. In the presence of the carnivorous plant, the behavioral and morphological changes of *C. dubia* are expressed simultaneously. The morphological changes alone may not suffice against a very effective predator like *U. x neglecta,* which can have a capture rate of 100% for undefended *C. dubia* in different juvenile instars [31]. In our predator avoidance experiments (see Appendix A), *Utricularia*-exposed *C. dubia* avoided the presence of *U. x neglecta* and *C. demersum* (Figure 4). Animals of the control group only avoided *U. x neglecta*. Potentially, *Utricularia*-exposed animals show higher alertness that makes them avoid any regions shaded by plants. This might be an alteration in phototactic behavior, as only our treatments that directly faced *U. x neglecta* or were exposed to it prior to the experiment showed significant ‘open water’ preferences. Control animals showed no significant avoidance of shaded areas. Fish evoke similar, but opposite behavioral responses in *D. magna*: Lauridsen and Lodge [53] demonstrated that *D. magna* seeks shelter in plant thickets when threatened by young sunfish (*Lepomis cyanellus*).

In our analysis of swimming modes and speed, we found that the ‘hop and sink’ mode, which is a less directed, slower movement, significantly increased in induced animals. Additionally, we found significantly reduced velocities of both observed modes in the *Utricularia*-exposed treatments compared to the control treatments. This overall reduction in swimming speed will either reduce the encounter rate between predator and prey [54], and/or reduce the possibility to activate the trigger hairs on the *U. x neglecta* trap door by reducing the kinetic energy of the animals [35]. Such a behavioral adaptation is also known from *D. magna*, who reduce their swimming velocity in the presence of fish cues or homogenized conspecifics [27,55]. A reduced swimming speed often comes at the cost of reduced feeding, which eventually leads to a reduced growth and fecundity [54].

### 3.4. Predation Trials

In our predation trials, we tested whether the above-described defenses are beneficial and render *C. dubia* less susceptible to this plant predator. We showed that induced animals expressing behavioral and morphological defenses are less often captured, and thus are better protected against *U. x neglecta.* As these phenotypic changes increase the survival of *C. dubia*, we hypothesize that they evolved in response to *U. x neglecta* predation. The increase of the survival rate of 7.5% in induced animals may seem rather insignificant on first sight, but it means *Utricularia* catching only every 40th daphniid instead of every 10th. Additionally, it is safe to assume that we only tested the early defensive effect in ontogeny, as these defensive structures grow even more pronounced over subsequent molts, as described for several daphniid species (e.g., [15]). 

### 3.5. Origin of the Defense-Inducing Stimulus

The origin of the cue that induces the observed alterations in *C. dubia’s* morphology, life-history, and behavior is unclear. Based on our experiments, we cannot exclude that *C. dubia* could sense *U. x neglecta* trap firings via mechano-receptors or identify the plant optically. However, we suggest that *C. dubia* detects chemical substances released by *U. x neglecta*. Since *U. x neglecta* and *C. dubia* were separated by net cages in our experiments, mechanical and visual cues were strongly damped, while chemical cues were not. Moreover, there are many examples described, especially in *Daphnia*, where predator presence is detected chemically [56,57]. It is known that they react to predator kairomones, but also to broadly defined alarm signals [58]. Alarm cues appear unlikely in our case given that prey organisms are not wounded during ingestion. Furthermore, we found reactions of *C. dubia* not only in fed *U. x neglecta* treatments, but also in unfed *U. x neglecta* treatments. This suggests that it is not an alarm cue from conspecifics, but a chemically active substance, a kairomone [59], released by *Utricularia*, but not directly connected to predation activity. In contrast to this, the kairomone of *Chaoborus* larvae is released with digestive liquids [8,60] and only induces neckteeth formation in *D. pulex* [61] if predators are feeding. Nonetheless, *U. x neglecta* fed with conspecifics of the investigated *C. dubia* induced stronger responses, e.g., a stronger reduction of body length (Figure 1). This suggests that the cue is stronger with successful capture or, at least, higher trap activity. For arming the traps, *U. x neglecta* bladders constantly pump water out of their interiors [32] (for which the mechanism and pathway are not yet fully understood). They also exhibit spontaneous firings once a critical negative pressure is achieved [62]. Moreover, prey capture leads to increased plant growth and the production of larger traps [63] with a higher spontaneous firing rate (and thus resetting rate) [64]. If *C. dubia* is able to sense (spontaneous) trap firings and detect the processes of digestion [65], trap resetting, respiration rate [66], and/or water excretion, *C. dubia* would have indirect measure(s) not only of trap presence, but also activity. 

## 4. Materials and Methods

### 4.1. Study Design

In order to depict a naturally occurring predator–prey system, we started this study by identifying local ponds containing *Utricularia x neglecta* (sensu 1 [67]) alongside several Daphniidae species in the field. We subsequently performed a trap analysis to validate the co-occurring Daphniidae species as prey items of *U. x neglecta*, as also reported in the literature (see Section 1 Introduction). From the resulting prey spectrum analysis [31], we chose *C. dubia* as our candidate for the present study due to its high abundance in the pond as well as in the *U. x neglecta* traps. For validating our initial hypothesis that *C. dubia* has evolved inducible defenses against the coexisting bladderwort (*U. x neglecta*), we adjusted the controlled laboratory experiments initially developed for animal predators (e.g., [10,12,13,19,31]). Based on our experience with Daphniidae and their inducible defenses, we aimed for a sample size of 10 specimens for each experiment, as we expected any alterations to be detectable with this sample size (e.g., [10,12,19,40,68]). The first experiment was designed to verify whether *C. dubia* reacts to the presence of *U. x neglecta* with alterations in morphology and life history. Using light microscopy, we measured morphometric (body length and body width) as well as life-history parameters (number of egg-carrying females, clutch size) of initially juvenile *C. dubia* specimen in four different treatments (tap water control, non-threatening plant, fed *Utricularia*, unfed *Utricularia*) over a duration of 6 days. Based on the initial findings, we conducted follow-up experiments in order to identify behavioral alterations as well as to validate the alterations as being an effective defense to the bladderwort traps. All experiments are described in detail below. We did not exclude any data from the analysis and outliers where not predefined or treated differently in the analysis. Randomization, where conducted, was used to prevent the influence of external factors (i.e., illumination), and no specific method for randomization was applied. Our study does not include any mode of blinding. 

### 4.2. Cultures

#### 4.2.1. Prey Crustaceans (Ceriodaphnia Dubia)

From the Gelsenkirchen pond samples, a clonal line of *C. dubia* (S04) was reared from a single female. This female and the subsequent offspring were cultured in 1 L beakers (J. Weck GmbH and Co. KG, Wehr-Öflingen, Germany) containing charcoal-filtered tap water. A maximum of 100 animals were kept in the beakers by transferring supernumerary adults and neonates into new beakers. The beakers were regularly cleared of detritus, half of the water was exchanged monthly, and *Acutodesmus obliquus* was added as a food source ad libitum. The cultures were kept under stable conditions at 20 °C ± 1 °C and a 16 h:8 h light to dark cycle.

#### 4.2.2. Predator (Carnivorous Plant *Utricularia x Neglecta*)

We used *U. x neglecta* initially purchased at Gartenbau Thomas Carow (Gartenbau Thomas Carow, Nüdlingen, Germany) and cultivated and used in prior experiments at the Botanical Garden of the University of Freiburg. For the experiments presented here, the plants were cultivated in the Department of Animal Ecology, Evolution and Biodiversity of the Ruhr-University Bochum, Germany. Plants were kept in 50 L plastic aquaria filled with charcoal-filtered tap water and positioned 60 cm beneath a light source consisting of four fluorescent tube lamps with 36 W each (Radium NL 36 W/840 Spectralux Plus cool white). The *U. x neglecta* culture was kept under the same stable conditions as the *C. dubia* culture (at 20 °C ± 1 °C and a 16 h:8 h light to dark cycle), and the plants were constantly growing and continuously producing new traps. The experimental research on plants complied with relevant institutional, national, and international guidelines and legislation.

### 4.3. Trap Entrance Dimensions

To measure the predator’s gape size, twenty *U. x neglecta* traps were dissected from the plant and imaged using a stereomicroscope (Olympus SZX16, Olympus Europa SE & Co. KG, Hamburg, Germany) with a digital camera (ColorView III digital imaging system) attached. The widths and heights of the trap entrances were measured via imaging software (Cell^D; Soft Imaging Solutions, SIS Olympus, Münster, Germany). As trap entrance width, we defined the shortest distance between opposite trap entrance walls, parallel to the threshold of the trap entrance margin [69]. The height of the trap entrance is the line connecting threshold and trap door insertion and is therefore orthogonal to the width.

### 4.4. Defense Induction

In order to investigate the *U. x neglecta*-induced morphological and life-history defenses in *C. dubia*, we analyzed individuals from the earliest juvenile stages. To do so, we started the experiments with egg-carrying individuals in the last embryonic stage and measured the offspring individually every 24 h throughout the following 6 days. We chose this ontogenetic stage because *Daphnia* is sensitive to predatory cues from the fourth embryonic stage onwards [10]. We conducted the experiment in a full factorial design consisting of four different treatments (*n* = 10 each). We used two different treatments in order to control for the absence of plants (‘tap water control’) as well as for the presence of non-threatening plants by exposing *C. dubia* to an equal amount of coontails (*Ceratophyllum demersum*) to the amount of *U. x neglecta* used in the experimental treatments (see below) (‘*Ceratophyllum* control’). Coontails naturally occur together with *U. x neglecta* [70] and *C. dubia*. As experimental treatments, we conducted two induction setups where *C. dubia* was confronted with *U. x neglecta.* In order to identify whether the biological activity is solely plant-borne, we reared *C. dubia* together with bladderworts as one experimental treatment (‘unfed *Utricularia*’). In addition, we performed an experimental treatment in which *C. dubia* was exposed to bladderwort that were fed daily with 25 juvenile *C. dubia* (‘fed *Utricularia*’), as inducing agents are often associated with active feeding processes [56,71]. All treatments were conducted in 1 L beakers (J. Weck GmbH and Co. KG, Wehr-Öflingen, Germany). To avoid direct predator contact and to prevent the consumption of the test specimens in both predator treatments, we separated the prey (*C. dubia*) and predator (*U. x neglecta*) using net cages equipped with fine mesh widths of 125 µm (Hydrobios, Altenholz, Germany). Within the net cages, we placed the egg-carrying *C. dubia* females. Plants (one shoot of 10–15 cm each) were placed outside the net cages and, depending on the treatment, were fed daily with 25 juvenile *C. dubia* (‘fed *Utricularia*’) or left unfed.

### 4.5. Analysis of Morphology and Life-History Alterations

In this initial experiment, we used four different treatments (‘tap water control’, ‘*Ceratophyllum* control’, ‘unfed *Utricularia’* and ‘fed *Utricularia’*; see Appendix A). Once the age-synchronized *C. dubia* females released their brood (i.e., approximately within 24 h), we removed the mothers and started to image the offspring in a daily rhythm for 6 days in total using a stereomicroscope equipped with a digital camera (ColorView III digital imaging system) and imaging software (Cell^D; Soft Imaging Solutions, SIS Olympus, Münster, Germany). We measured body length, body width, the number of egg-carrying females, and the average number of eggs deposited in the brood pouch. The body length was measured from the top of the compound eye to the point where the carapace converges into the tail-spine. Body width was measured at the broadest distance between ventral and dorsal perpendicular to the body length. In order to analyze the body width allometrically, we normalized it to the body length (normalized body width = body width/body length).

### 4.6. 3D Analysis of Morphological Alterations

In order to identify the morphological alterations comprehensively, we conducted a three-dimensional analysis of the control and plant-exposed *C. dubia*. For that, we used *C. dubia* (*n* (control) = 13; *n* (induced) = 8) individuals from the ‘fed *Utricularia*’ treatment on day 5 of the experimental period. The animals were stained using Congo red, scanned on a confocal laser scanning microscope, and subsequently digitized as a surface image. These surface images were analyzed using a landmark-based method (≈45.000 semi-landmarks per animal) and compared using a Procrustes-based analysis. For details, please see Horstmann et al. [40].

### 4.7. Analysis of Behavioral Defenses

#### 4.7.1. Predator Avoidance

We conducted a subsequent experiment that aimed to identify behavioral changes in *C. dubia* as a response to the presence of *U. x neglecta*. We designed this experiment in order to test whether *C. dubia* avoids areas that are shadowed by plants depending on their stage of alertness (either naïve or alerted by prior predator exposure). We used five different setups resulting from the combination of two different treatments (‘control’ and ‘fed *Utricularia*’) and three different experimental scenarios (‘no plant’, ‘*Elodea*’, and ‘*Utricularia*’). The combination ‘fed *Utricularia*’/‘no plant’ was not included in our experiments. As specimens, we used five-day-old *C. dubia* that were either reared in ‘control’ or in ‘fed *Utricularia*’ beakers. The ‘no plant’ scenario with ‘control’ animals was used as behavioral baseline (‘tap water control’). The avoidance behavior of the two treatments was tested in the two environmental scenarios: a control condition with the non-carnivorous plant *Elodea canadensis*, and a test treatment with the carnivorous plant *U. x neglecta*. We wanted to test for external factors affecting behavior (e.g., inhomogeneous light conditions, as a result of plant associated shading) and used exposure to *Elodea canadensis* as a comparison to the control condition without any plants, because *E. canadensis* shows strong similarity to *U. x neglecta* in terms of shadowing. Their color and whorl morphology give them an *Utricularia*-like appearance. The plant treatments were conducted using a single shoot of *U. x neglecta* or *E. canadensis*, respectively. For each treatment, we placed 20 five-day-old *C. dubia* in 2 L plastic tanks (ca. 18 cm × 13 cm × 11.5 cm, Savic, Heule Kortrijk, Belgium) filled with charcoal-filtered tap water and, according to the experimental conditions, *U. x neglecta* or *E. canadensis* randomly positioned on either side of the tank. The plants were kept on one side of the tank with a spacer positioned centrally in the respective tank, fixing the floating plants. All five different experimental setups (‘tap water control’, ‘control vs. *Elodea*’, ‘fed *Utricularia* vs. *Elodea*’, ‘control vs. *Utricularia*’, ‘fed *Utricularia* vs. *Utricularia*’) were started simultaneously and were monitored in parallel. The experiment was repeated 10 times. For the documentation of the animals’ positions, the tanks were divided into 18 equally sized sections (each approx. 3 × 4.3 cm) by superimposing a grid with three rows and six columns on the tanks’ fronts. Three of these columns did not contain plants, and three columns contained plants. For homogeneous light conditions and to avoid light reflections, we installed a single fluorescent tube lamp above each tank (fluorescent tube lamp, Radium NL 36 W/840 Spectralux Plus cool white). This setup provided uniform light over the whole surface and prevented shadows. Furthermore, the treatments were randomly permutated between the tanks in order to exclude position-dependent effects (e.g., whether there were neighboring tanks or not). We started the experiment by introducing the 20 five-day-old *C. dubia* after acclimation for 30 min to the new environment, as used in comparable studies [27,72]. We manually documented the distribution pattern of *C. dubia* in the sections of the tank every 15 min for a total duration of 60 min, resulting in five measurements for each treatment (0 min, 15 min, 30 min, 45 min, 60 min). The animal distribution data were tested for differences over time within each treatment. The respective ANOVAs that tested every treatment for differences between the subsequent measurements did not reveal any significant differences, and the data were therefore pooled for each treatment over time. 

#### 4.7.2. Swimming Velocity

To determine adaptive swimming behavior, we used ‘control’ and ‘fed *Utricularia*’ specimens. After preparing the treatments, the individuals were placed into a tank (12.5 cm × 10 cm × 2.5 cm) containing only charcoal-filtered tap water (20 °C ± 1 °C) and were given five minutes for acclimation before the recordings began [27]. We recorded the animals for five minutes at a frame rate of 30 fps using a Nikon D5100 (equipped with Nikon DX AF-S Nikkor 18–105 mm 1:3.5–5.6 G ED; Nikon Corporation, Tokyo, Japan). Afterwards, we analyzed ca. 800 sequences of 5 s in which the animals were moving in a straight line and in parallel to the tank’s front pane. Movement of the animals’ geometric centers were tracked by hand using a self-scripted MATLAB application (MATLAB R2014b, The Mathworks Inc., Natick, MA, USA, 2015). This program delivers the swimming velocity at any point in time and was subsequently used to calculate an average velocity for each individual. In total, we recorded and analyzed the swimming movements of 200 animals of each treatment.

#### 4.7.3. Swimming Mode

In *Daphnia*, three different swimming modes can be classified: ‘hop and sink’, ‘zooming’, and ‘looping/spinning’ [50,73,74,75,76]. The ‘hop and sink’ mode is characterized by alternating upward movements, powered by forceful strokes of the second antennae (hops), interrupted by periodical breaks (sink). In the ‘zooming’ mode, daphniids display a series of fast swimming strokes with no sinking phases in between. In comparison, the ‘hop and sink’ mode is a rather slow swimming mode (<10 mm/s), whereas the ‘zooming’ mode is rather fast (>15 mm/s) [74]. The ‘looping/spinning’ mode is displayed as a series of backward loopings. From the aforementioned recorded videos, we randomly analyzed 65 videos per treatment and determined the proportions of the swimming modes ‘hop and sink’ and ‘zooming’, since these were the dominant movement patterns. This was done by randomly choosing a time frame of 30 s in each of these videos, in which the animal was clearly visible and swimming in parallel to the tank’s front pane. 

### 4.8. Predation Experiments

We conducted predation trials to determine the effect of phenotypic changes on *U. x neglecta* capture efficiency. For that, we placed 20 five-day-old animals raised in ‘fed *Utricularia*’ or ‘control’ treatments into a glass vial filled with 40 mL of charcoal-filtered tap water that contained a 5 cm long shoot of *U. x neglecta* possessing 30 empty traps. This setup was placed in a climate chamber at 20 °C ± 1 °C and a day–night cycle of 16:8 h for 24 h, and afterwards, the surviving rate was determined. Ten replicates were conducted for each treatment. 

### 4.9. Statistical Analysis

For the statistical analysis of our experimental data, we used R x64 3.4.2 [77] with a significance threshold ≤ 0.05 for all conducted tests. The packages “ggplot2” [78], “gdata” [79], “ggpubr” [80], “ggsignif” [81], and “rstatix” [82] were used for plots and tests.

The data of the 2D measurements followed a normal distribution (Shapiro test), so we conducted a multivariate analysis of variance (MANOVA) with post hoc test (Bonferroni-corrected pairwise *t* test) to compare the four treatments across the six consecutive days of the experiment. We calculated η^2^ to estimate effect sizes based on the model used for the MANOVA.

The data of the life-history parameters were analyzed using a Kruskal–Wallis rank sum test (Bonferroni-corrected pairwise Wilcoxon test) followed by the determination of effect sizes using η^2^ for each day.

The 3D data were based on the computed comparisons of the averaged point positions, using a displacement vector approach and, furthermore, the point translocations along the coordinate axes (refer to [40] for details). We tested these axes-wise point translocations with Wilcoxon tests at a significance level ≤ 0.01, conducted within the MATLAB environment. Significance levels were adjusted for multiple testing based using the false discovery rate (FDR) approach [83]. This approach estimates the probability of declaring a not-differing feature as significantly different among all significant features, given as a ‘q-value’. Finally, the 3D forms of the plant-exposed and control individuals were compared using confidence ellipsoids. We calculated the effect size Pearson’s *r* using R for each conducted Wilcoxon test and averaged them (mean) for each analyzed axis.

For the statistical analysis of swimming velocity and swimming mode, we conducted Kruskal–Wallis rank sum tests followed by Bonferroni-corrected pairwise Wilcoxon tests between the respective treatments. Finally, we calculated η^2^ to determine the effect sizes.

The predation experiment data did not follow a normal distribution and was therefore analyzed using non-parametric methods. The treatments were tested for differences using a Mann–Whitney U test followed by a calculation of Pearson’s *r* for effect size.

## 5. Conclusions

Predator-induced phenotypic plasticity is discussed to evolve under certain circumstances [4]. First, the predation pressure must be variable and occasionally strong. Second, the predator must be perceptible by a reliable cue. Third, the induced defense must be effective. Fourth, the defense should be associated with costs or trade-offs. *U. x neglecta* shows variability in abundance throughout the year, with high abundances during summer and a resting stage during winter [84]. Furthermore, the trap number of *U. x neglecta* varies according to biotic and abiotic factors, reaching peak densities that pose a severe threat to zooplankters [85]. Given that *U. x neglecta* exhibits the necessary variability in trap abundance and causes high predatory pressure at least during the summer months, the first prerequisite for inducible defenses is already fulfilled. Second, we present strong evidence for a reasonably reliable cue that enables *C. dubia* to perceive *U. x neglecta* and react on its presence with a set of behavioral and morphological alterations. Third, we show that these adaptive changes are effective, as induced *C. dubia* are consumed less by *U. x neglecta*. Fourth, our experiments also show that the fecundity of induced animals is reduced, thus, these alterations come at the expense of the population growth rate. In summary, our study strongly suggests the evolution of animal-inducible defenses against a predatory plant.

With inducible defense strategies often being highly predator-specific, and the fact that *U. x neglecta* is only one representative of a cosmopolitan genus containing more than 250 species, we expect that *C. dubia* is not the only member of the Daphniidae family to thwart this ‘green threat’ with inducible defenses. The carnivorous waterwheel plant (*Aldrovanda vesiculosa*) with snap-traps is another aquatic predator for daphniids and other zooplankters [86,87]. In fact, given the variety of carnivorous plants, their trapping principles, and sometimes narrow prey spectra [88], there are probably a number of inducible defenses against them yet to be identified in different species and ecosystems.

## Figures and Tables

**Figure 1 ijms-23-06474-f001:**
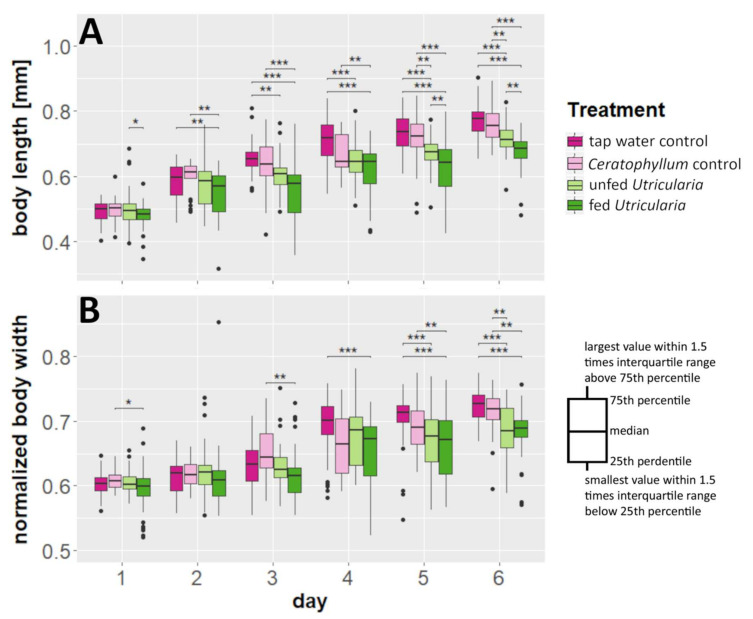
Morphological changes in *C. dubia* as a response to the presence of *U. x neglecta*. (**A**) Body length measurements over a duration of 6 days for four different treatments including two control treatments and two *Utricularia*-exposed treatments. (**B**) Normalized body widths (body width/body length). *Utricularia*-exposed animals show significantly smaller body length and normalized body width than the control treatments. * *p* < 0.05; ** *p* < 0.01; *** *p* < 0.001.

**Figure 2 ijms-23-06474-f002:**
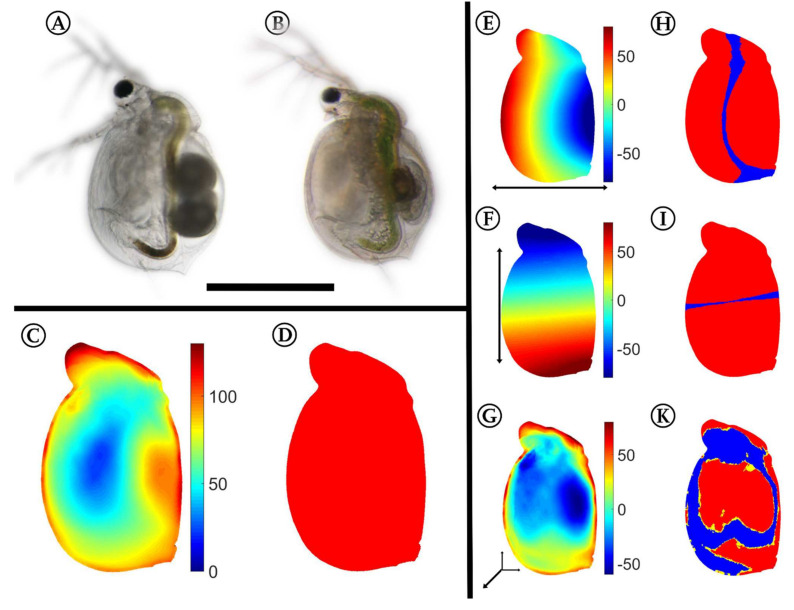
3D analysis of morphological alterations between control and *Utricularia*-exposed *C. dubia*. Control (**A**) and *Utricularia*-exposed *C. dubia* (**B**) of same age, scale bar = 1 mm. All subsequent analyses are projected on the average *Utricularia*-exposed animal. (**C**) Overall deformation; strong shifts are colored in shades of red, while small or no changes are indicated by shades of blue. (**D**) Confidence ellipsoid plot, revealing no overlapping confidence ellipsoids. (**E**,**F**) Here, shades of red indicate a shift in positive direction on that axis (dorsal/anterior/distal), shades of blue indicate a shift in negative direction on the respective axis (ventral/posterior/proximal). Shifts along the anterior–posterior (**E**) and dorso–ventral axis (**F**) show that the animals are smaller in the *Utricularia*-exposed morph. The deformation in the lateral dimension (**G**) gives regions of reduced and increased body width. Most of the found shifts are proven significant with respective Wilcoxon tests and FDR-based q-values (**H**,**I**,**K**). These figures give regions with *p*-values of respective Wilcoxon tests lower than 0.01 colored yellow, regions that also showed q-values lower than 0.001 are colored red. For the respective analysis, all samples of both treatments were taken into account (n_induced_ = 8, n_control_ = 13).

**Figure 3 ijms-23-06474-f003:**
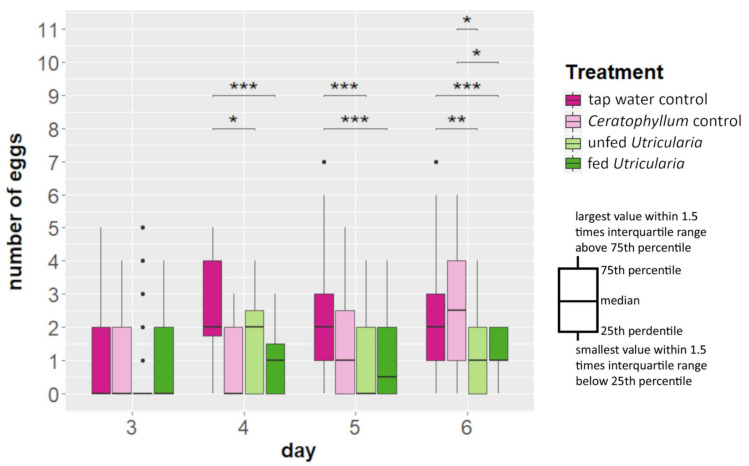
Changes in life history (in terms of clutch size alterations) of *C. dubia* in the presence of *U. x neglecta*. *C. dubia* revealed smaller clutch sizes (*p* ≤ 0.01) in the presence of *U. x neglecta* compared to the control treatments from day 4 onwards, stagnating at about one egg per female. * *p* < 0.05; ** *p* < 0.01; *** *p* < 0.001.

**Figure 4 ijms-23-06474-f004:**
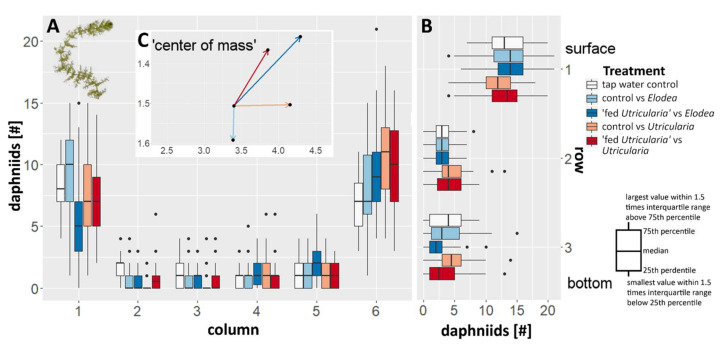
Changes in behavior in *C. dubia* observed as averaged distribution pattern with respect to the presence of either *U. x neglecta* or *E. canadensis*. (**A**) The box plots indicate the number of animals per column in the canvas drawn on the tank front pane. Increasing numbers on the x axis are equal to an increase in distance to the respective plant used in that treatment (1 equals to the same column as the plant, 6 is the opposite tank side). (**B**) The box plots indicate the number of animals per row. (**C**) The additional vector graph inlet is indicating the average positioning of the animals in respect to the plant by depicting a vector that represents the ‘calculated center of mass’ for every treatment as an offset from the tap water control treatment.

**Figure 5 ijms-23-06474-f005:**
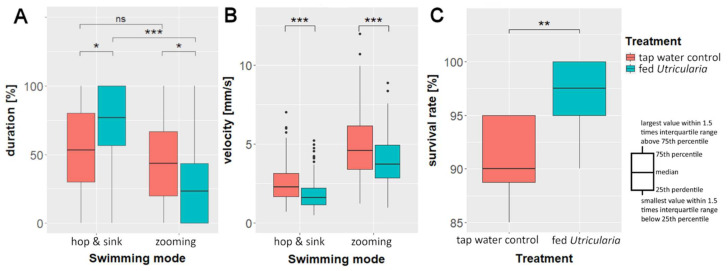
Behavioral changes in *C. dubia* as response to the presence of *U. x neglecta*: average duration (**A**) and velocity (**B**) of the two observed swimming modes in the ‘swimming modes’ experiments. (**C**) Survival rate of 20 five-day-old *C. dubia* (either control or *Utricularia*-exposed) over 24 h in the presence of 30 *U. x neglecta* traps. * *p* < 0.05; ** *p* < 0.01; *** *p* < 0.001.

## Data Availability

All data are included in the manuscript and Appendix A.

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
