# Peer review of "Facing the Green Threat: A Water Flea’s Defenses against a Carnivorous Plant"

_ijms, 2022, doi:10.3390/ijms23126474_

Round 1

Reviewer 1 Report

Review ID ijms-1753049

 Facing the green threat: A waterflea’s defenses against a carnivorous plant

               This is quite well organized manuscript. I found this “ms” interesting and innovative. However, a few questions must be explained more precisely.

Critical review:

1.     This manuscript looks very interesting, even I am a bit far from the topic of the study.

2.     Introduction should be more precisely described. It looks quite poor. Are they no more papers available treating the same study or similar one?

3.     The aim of the study is not well explained.

4.     Well, I always thought conclusion is defined as a heart of the story. It doesn't look like. 2-3 sentences are of importance for the readers. More essential and innovative information in the Conclusion section is needed.

5.     It is of importance to mention the natural plant's defence strategy, where volatiles play a crucial role. Some papers below. Carnivorous or not but this is a plant, isn't it?

Some other paper to add:

Tribolium confusum responses to blends of cereal kernels and plant volatiles

Journal of Applied Entomology 140, 558–563 (2016)

DOI: 10.1111/JEN.12284

Sitophilus granarius responses to blends of five groups of cereal kernels and one group of plant volatiles

Journal of Stored Products Research 62: 36-39 (2015)

DOI: 10.1016/J.JSPR.2015.03.007

Effect of phenolic acid content on acceptance of hazel cultivars by filbert aphid

Plant Protection Science 55: 116-122 (2019)

DOI: 10.17221/150/2017-PPS

Author Response

Dear reviewer,

we appreciate your positive response to our manuscript and provide a point by point response to your review below.

  1. We appreciate your assessment of our study.
  2. We are not aware of similar studies except the one we cited (Havel and Dodson 1985). We now point that out more clearly in the introduction (l. 56ff).
  3. We now phrase our study's lead questions in order to better explain our aims. l. 73ff
  4. In our opinion, the conclusion indeed boils down our findings to the "heart of the story". In fact, we connect the findings of our study to the commonly accepted prerequisites for the evolution of inducible defenses. By doing so, we anchor our conclusions in the ecological framework. Furthermore, we provide an assessment about ecosystems that may include similar phenomena. Maybe we missed the point the reviewer raised about our conclusion but we do not see the conclusion being deficient even by the reviewers definition.
  5. We'd like to thank the reviewer for bringing up this ecologically relevant topic. We acknowledge the importance of plant defensive volatiles and their consequences in ecosystems. That is why we mention them in our discussion (l. 222ff). However, we do not see the urge to include respective information into our introduction as we describe a very different plant-animal interaction in which which the animals defend against the plant. Furthermore, the plant would benefit from attracting rather than repelling grazers. Plant defensive volatiles are well studied and commonly known amongst the ecological science community. Hence, including respective information appears gratuitous and would further increase the volume of our already extensive manuscript.

Best regards,

Sebastian Kruppert, corresponding author

Reviewer 2 Report

Manuscript Number: ijms-1753049, titled:

 Facing the green threat: A waterflea’s defenses against a carnivorous plant

Review 1 – 25 May 2022

To Authors:

  1. Introduction section, line 38 and in the whole manuscript, please, verify how to insert the references numbers. Use some recently published paper as a sample;
  2. In the whole manuscript, after the Genus and the species, please, insert the abbreviation of the Botanist;
  3. 4.1 sub-section, separate 4.1 from Study design;
  4. Verify id the captions of figures are consistent with the published papers of IJMS;
  5. References section: the references are not included as per IJMS (see ref 83) and the whole section;

I suggest a minor revision of this manuscript.

Regards.

Author Response

Dear reviewer,

thank you very much for spending all the effort to find all these formatting mistakes, we corrected them all and provide a point by point response below:

  1. We corrected our citation style according to to the MPDI standard.
  2. We included the Botanists abbreviation in the first occurence of the species name in the abstract as well as in the introduction.
  3. We separated "4.1" from "Study design", thanks for finding that formatting mistake.
  4. We formatted the figure captions to the journal's standard.
  5. We corrected our reference style according to the MPDI standard.

Best regards,

Sebastian Kruppert, corresponding author